# Detecting and Quantifying Structural Breaks in Climate

**Neil R. Ericsson [1,2,3,*], Mohammed H. I. Dore [4] and Hassan Butt [5]**

1. Division of International Finance, Board of Governors of the Federal Reserve System,
   Washington, DC 20551, USA
2. Department of Economics and H.O. Stekler Research Program on Forecasting, The George Washington
   University, Washington, DC 20052, USA
3. Paul H. Nitze School of Advanced International Studies (SAIS), Johns Hopkins University,
   Washington, DC 20036, USA
4. Department of Economics, Climate Change Laboratory, Brock University,
   St. Catharines, ON L2S 3A1, Canada
5. Department of Economics, Climate Change Laboratory, Brock University, St. Catharines, ON L2S 3A1, Canada
*  Correspondence: ericsson@frb.gov

**Abstract:** Structural breaks have attracted considerable attention recently, especially in light of the financial crisis, Great Recession, the COVID-19 pandemic, and war. While structural breaks pose significant econometric challenges, machine learning provides an incisive tool for detecting and quantifying breaks. The current paper presents a unified framework for analyzing breaks; and it implements that framework to test for and quantify changes in precipitation in Mauritania over 1919–1997. These tests detect a decline of one third in mean rainfall, starting around 1970. Because water is a scarce resource in Mauritania, this decline—with adverse consequences on food production—has potential economic and policy consequences.

**Keywords:** climate change; indicator saturation; machine learning; Mauritania; rainfall; structural breaks

## 1. Introduction

Structural breaks have attracted considerable attention over the last two decades, particularly in light of the financial crisis, the Great Recession, the COVID-19 pandemic, supply-chain bottlenecks, and war in Ukraine. From an econometric perspective, the empirical presence of structural breaks poses significant challenges to modeling, both in empirical model selection and in statistical inference. That said, recent developments in machine learning provide incisive tools for detecting and quantifying structural breaks. Empirically, the extent and nature of climate change—*qua* structural breaks—is of considerable interest and contention, cf. McShane and Wyner (2011). Moreover, structural breaks in climate can have economic, financial, political, and social ramifications, as highlighted in the United Nations (1992) Framework Convention on Climate Change (UNFCCC), the Paris Agreement at the 2015 UNFCCC Conference of the Parties (COP 21), and COP 27.

The current paper has two objectives. First, it presents a unified framework for the econometric analysis of structural breaks, drawing on the literature on indicator saturation techniques and automatic model selection with machine learning. Second, it applies those techniques to detect and quantify breaks in the pattern of precipitation within Mauritania—testing whether statistically and empirically significant changes have occurred and, if so, quantifying those changes.

The econometric evidence shows declining precipitation in Mauritania, starting around 1970, with a possible increase two decades later. Economically, the changing pattern of precipitation is likely to aggravate the persistent food deficit in Mauritania. One possible policy response would be a long-term plan for adaptation to the declining rainfall, including modification of farming practices. In such adaptation, Mauritania—as a developing

country—could rely on the provisions of the UNFCCC and might expect to receive financial assistance. In particular, the econometric evidence shows a structural break in Mauritanian rainfall that could form a basis for establishing a time threshold determining compensation for climate change.

This paper is organized as follows. Section 2 discusses various procedures for detecting structural change, focusing on impulse indicator saturation and several of its extensions. Section 3 reviews some background pertaining to Mauritania and the Sahel. Section 4 evaluates the Palmer Drought Severity Index (PDSI) for Mauritania, describes the underlying rainfall data available for Mauritania, and derives the aggregated measure of rainfall to be examined. Section 5 applies the procedures in Section 2 to the data on Mauritanian rainfall. Section 6 comments on the methodologies employed and the results obtained and considers directions for further research. Section 7 concludes.

## 2. Methodologies for Detecting Structural Breaks

This section discusses various common statistical techniques for detecting structural breaks, which Section 5 then uses to test for and quantify breaks in the pattern of rainfall in Mauritania. To provide a unifying framework, the current section's exposition focuses on impulse indicator saturation (IIS), integrating and building on expositions and developments in Ericsson (2011b, 2012, 2017a) and Ericsson and Reisman (2012). Sections 2.1 and 2.2, respectively, discuss IIS and extensions of IIS as procedures for model evaluation, including as tests for parameter constancy.

### 2.1. Impulse Indicator Saturation

This subsection summarizes how impulse indicator saturation provides a general procedure for analyzing a model's constancy. Specifically, IIS is a generic test for an unknown number of breaks, occurring at unknown times anywhere in the sample, with unknown duration, magnitude, and functional form. IIS is a powerful empirical tool for both evaluating and improving existing empirical models. Hendry (1999) proposes IIS as a procedure for testing parameter constancy. For further discussion and recent developments, see Hendry et al. (2008); Doornik (2008, 2009); Johansen and Nielsen (2009, 2013, 2016); Hendry and Santos (2010); Ericsson (2011a, 2011b, 2012, 2016); Ericsson and Reisman (2012); Hendry and Pretis (2013); Bergamelli and Urga (2014); Hendry and Doornik (2014); Castle et al. (2015); Marczak and Proietti (2016); Pretis et al. (2016); Castle et al. (2020), and Nielsen and Qian (2022).

Impulse indicator saturation uses the zero-one impulse indicator dummies to analyze properties of a model. For a sample of $T$ observations, there are $T$ such dummies, one for each observation in the sample. That is, for the $i$-th impulse indicator dummy $I_{it}$ ($i = 1, \ldots, T$), $I_{it} = 1$ for $t = i$, and $I_{it} = 0$ for all $t \neq i$ ($t = 1, \ldots, T$). Unrestricted inclusion of all $T$ dummies in an estimated model (thereby "saturating" the sample) is infeasible. However, blocks of dummies *can* be included, and that insight provides the basis for IIS. To motivate how IIS is implemented in practice, this subsection employs a bare-bones version of IIS in two simple Monte Carlo examples.

**Example 1.** *This example illustrates the behavior of IIS when the model is correctly specified. Suppose that the data generation process (DGP) for the variable $w_t$ is:*

$$w_t = \mu_0 + \varepsilon_t \quad \varepsilon_t \sim \mathrm{NID}(0, \sigma^2), \quad t = 1, \ldots, T, \tag{1}$$

*where $w_t$ is normally and independently distributed* (NID) *with mean $\mu_0$ and variance $\sigma^2$. Furthermore, suppose that the model estimated is a least-squares regression of $w_t$ on an intercept, i.e., the model is correctly specified. Figure 1a plots Monte Carlo data from a single replication of the DGP in Equation (1) with $\mu_0 = 20$, $\sigma^2 = 1$, and $T = 100$. Figure 1b plots the estimated model's residuals, scaled by that model's residual standard error.*

*The bare-bones version of IIS is as follows.*

1.  *Estimate the model, including impulse indicator dummies for the first half of the sample, as represented by Figure 2a. That estimation is equivalent to estimating the model over the second half of the sample, ignoring the first half. Drop all statistically insignificant impulse indicator dummies and retain the statistically significant dummies (Figure 2b).*
2.  *Repeat this process, but start by including impulse indicator dummies for the* second *half of the sample (Figure 2d), and retain the significant ones (Figure 2e).*
3.  *Re-estimate the original model, including all dummies retained in the two block searches from Steps #1 and #2 (Figure 2g), and select the statistically significant dummies from that combined set (Figure 2h).*

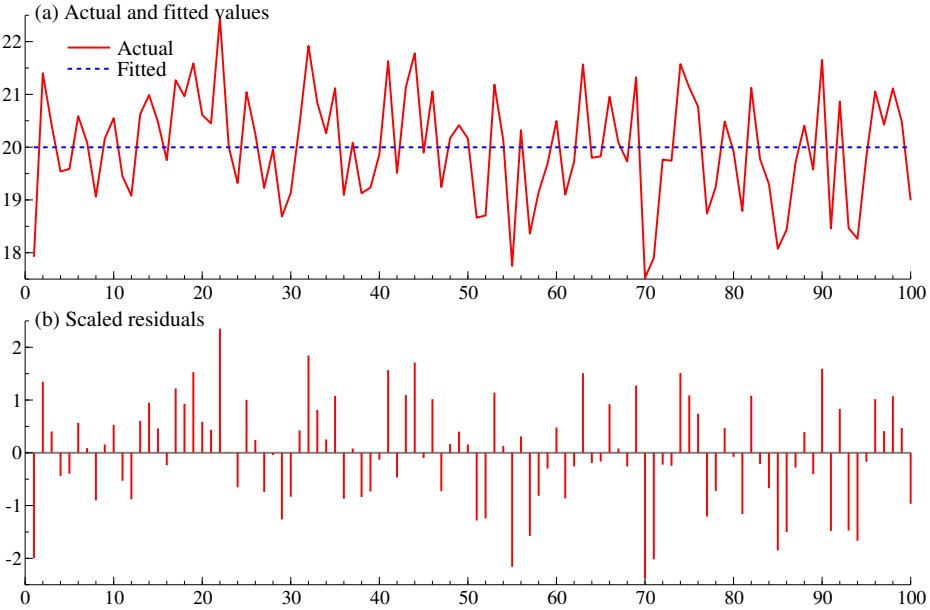

**Figure 1.** Actual and fitted values and the corresponding scaled residuals for the estimated model when the DGP does not have a break.

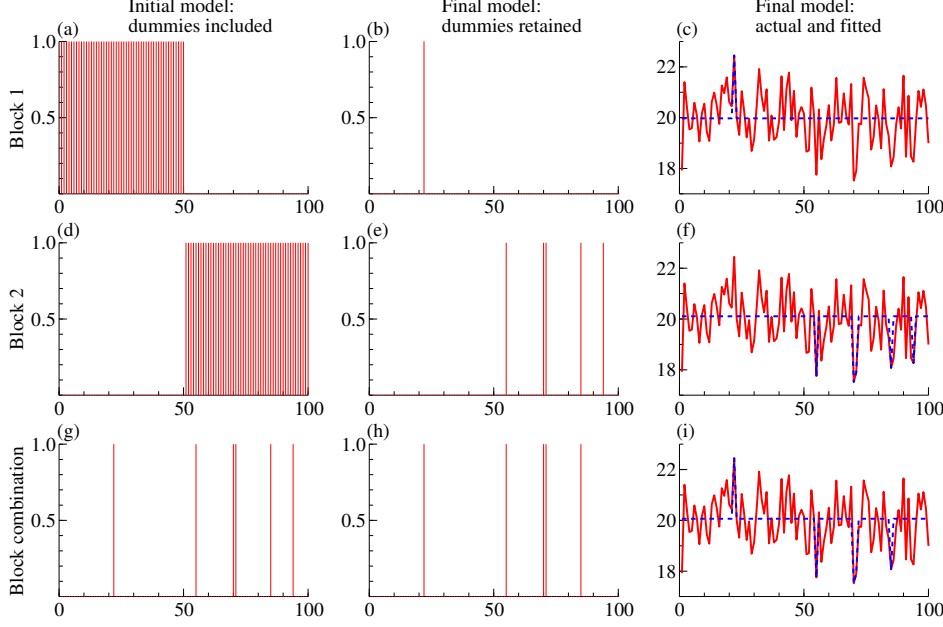

**Figure 2.** A characterization of bare-bones impulse indicator saturation with a target size of 5% when the DGP does not have a break.

Hendry et al. (2008) and Johansen and Nielsen (2009) have shown that, under the null hypothesis of correct specification, the expected number of impulse indicator dummies retained is roughly $\alpha T$, where $\alpha$ is the target size. In Figure 2h, five dummies are retained; $\alpha = 5\%$; and $\alpha T = (5\% \cdot 100) = 5$, an exact match.

**Example 2.** *This example illustrates the behavior of IIS when there is an unmodeled break, and hence the model is incorrectly specified. Suppose that the DGP for the variable $w_t$ is:*

$$w_t = \mu_0 + \mu_1 S_{64t} + \varepsilon_t \quad \varepsilon_t \sim \text{NID}(0, \sigma^2), \quad t = 1, \ldots, T, \tag{2}$$

*where $S_{64t}$ is a one-off step dummy that equals 0 ($t = 1, \ldots, 63$) or 1 ($t = 64, \ldots, 100$), and $\mu_1$ is its coefficient in the DGP. The model estimated is a least-squares regression of $w_t$ on an intercept alone, ignoring the break induced by the step dummy $S_{64t}$. As in Example 1, $w_t$ is normally and independently distributed with a nonzero mean. However, at $t = 64$, that mean alters from $\mu_0$ to $(\mu_0 + \mu_1)$. The model ignores that change in mean (aka a "location shift") and hence is mis-specified. Figure 3a plots Monte Carlo data from a single replication of the DGP in Equation (2) with $\mu_0 = 20$, $\mu_1 = -10$, $\sigma^2 = 1$, and $T = 100$. Figure 3b plots the estimated model's residuals. Interestingly, no residuals lie outside the estimated 95% confidence region, even though the break is $-10\sigma$. The model has no "outliers".*

*Figure 4 plots the corresponding graphs for the bare-bones implementation of IIS described in Example 1, as applied to the Monte Carlo data in Example 2. The block searches for Steps #1 and #2 retain many dummies. However, in the search across the combined retained dummies from those two searches, several irrelevant dummies are dropped. Furthermore, as the penultimate graph (Figure 4h) shows, the procedure has high power to detect the break, even although the nature of the break is not utilized in the procedure itself.*

In practice, IIS as an algorithm may be more complicated than this bare-bones version, which employs two equally sized blocks, selects dummies by $t$-tests, and is non-iterative. Doornik and Hendry's (2018) Autometrics econometrics software implements IIS in an algorithm with machine learning with several enhancements: IIS utilizes many possibly unequally sized blocks, rather than just two blocks; the search procedure iterates, and the partitioning of the sample into blocks may vary over iterations; dummy selection includes $F$-tests against a general model; and residual diagnostics help guide model selection. See also Hendry and Krolzig (1999; 2001; 2005); Hoover and Perez (1999, 2004); and Krolzig and Hendry (2001). Notably, the specific algorithm for IIS can make or break IIS's usefulness; cf. Doornik (2009); Castle et al. (2010); and Hendry and Doornik (2014). IIS is a statistically valid procedure for integrated, cointegrated data; see Johansen and Nielsen (2009). IIS can serve as a diagnostic statistic for many forms of mis-specification, and it can aid in model development.

Many existing procedures can be interpreted as special cases of IIS in that they represent particular algorithmic implementations of IIS. As Table 1 summarizes, such special cases include Fisher's (1922) covariance statistic, recursive estimation, the Chow (1960) predictive failure statistic (including the one-step, breakpoint, and forecast versions implemented in OxMetrics), the unknown breakpoint test (proposed by Nyblom (1989); Hansen (1992); and Andrews (1993)); and the Bai and Perron (1998) multiple breakpoint test. IIS also includes rolling regression, the tests of extended constancy in Ericsson et al. (1998, p. 305ff), tests of nonlinearity, intercept correction (in forecasting), and robust estimation. IIS thus provides a general and generic procedure for analyzing a model's constancy, allowing for an unknown number of structural breaks occurring at unknown times with unknown duration and magnitude anywhere in the sample.

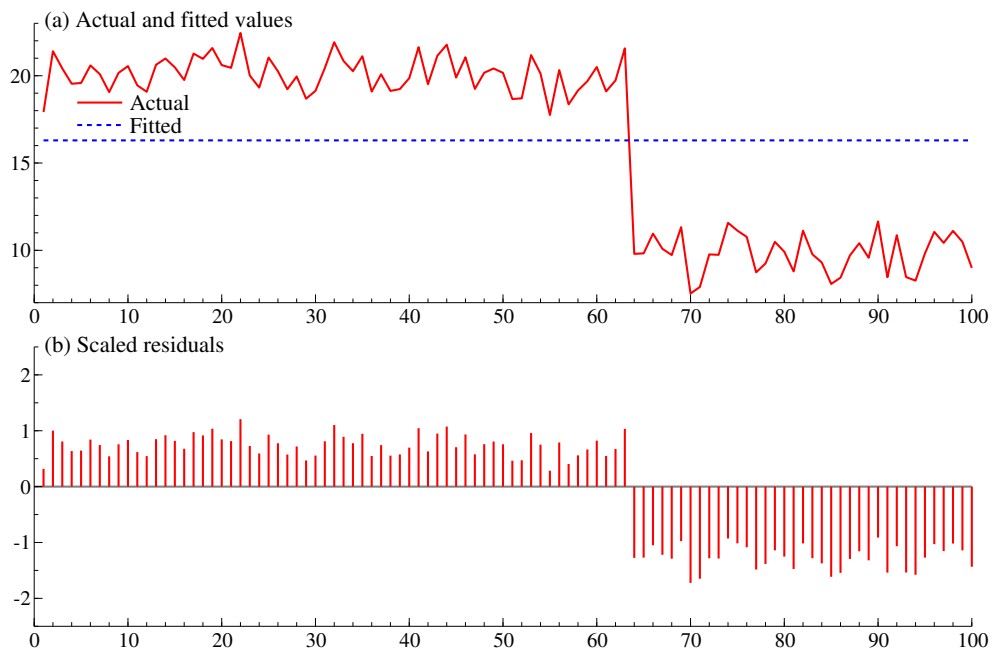

**Figure 3.** Actual and fitted values and the corresponding scaled residuals for the estimated model when the DGP has a break and the model ignores that break.

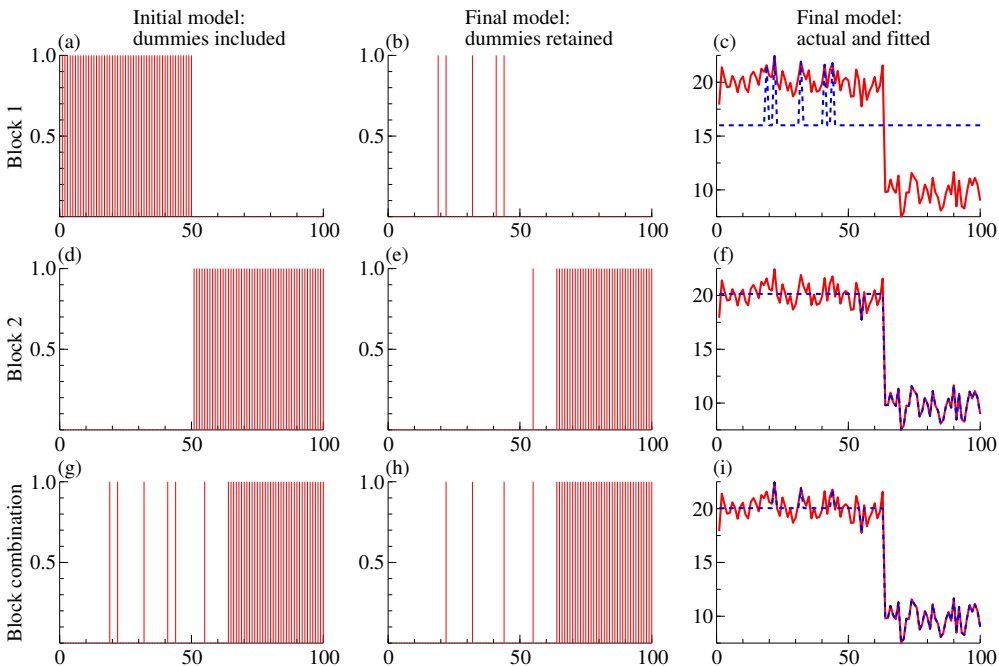

**Figure 4.** A characterization of bare-bones impulse indicator saturation with a target size of 5% when the DGP has a break and the model ignores that break.

## 2.2. Extensions

Table 1 includes several extensions of IIS. For ease of discussion of these extensions, consider IIS as a reference. Throughout, $T$ is the sample size, $t$ is the index for time, $i$ and $j$ are the indexes for indicators, $k$ is the index for economic variables (denoted $x_{kt}$), and $K$ is the total number of potential regressors considered.

**Table 1.** Selected developments in tests of structural breaks.

| Focus of Test Statistic | References |
|---|---|
| Covariance equality | Fisher (1922) |
| Recursive estimates | Plackett (1950) |
| Forecast errors | Chow (1960) |
| Single unknown breakpoint in regression coefficients | Nyblom (1989); Andrews (1993), Hansen (1992) |
| Multiple unknown breakpoints in the intercept | Bai and Perron (1998) |
| Arbitrary unknown impulse breaks (impulse indicator saturation) | Hendry (1999), Hendry et al. (2008), Johansen and Nielsen (2009) |
| Arbitrary unknown step breaks (step indicator saturation) | Castle et al. (2015) |
| Step, trend, and coefficient breaks (super saturation, ultra saturation, and multi-saturation) | Ericsson (2011b) |

*Impulse indicator saturation.* This is the standard IIS procedure proposed by Hendry (1999), with selection among the $T$ zero-one impulse indicators $\{I_{it}\}$.

*Super saturation.* In super saturation, searches are across all possible one-off step functions $\{S_{it}\}$, in addition to all impulse dummies $\{I_{it}\}$. Step functions are of economic interest because they may capture permanent or long-lasting changes that are not otherwise incorporated into a specific empirical model. A step function is a partial sum of impulse indicators. Equivalently, a step function is a parsimonious representation of a sequential subset of impulse indicators that have equal coefficients. Castle et al. (2015) investigate the statistical properties of a closely related saturation estimator—step indicator saturation (SIS)—which searches among only the step indicator variables $\{S_{it}\}$. Nielsen and Qian (2022) derive asymptotic properties of the gauge of SIS.

*Ultra saturation.* Ultra saturation (earlier, sometimes called "super duper" saturation) searches across $\{I_{it}, S_{it}, T_{it}\}$, where the $\{T_{it}\}$ are broken linear trends. Broken linear trends may be of economic interest. Mathematically, the $\{T_{it}\}$ are partial sums of the partial sums of impulse indicators. Broken quadratic trends, broken cubic trends, and higher-order broken trends are also feasible.

*Other saturation-based approaches.* Table 1 is by no means an exhaustive list of extensions to IIS. Other extensions include sequential ($j = 1$) and non-sequential ($j > 1$) pairwise impulse indicator saturation for an indicator $P_{it}$, defined as $I_{it} + I_{i+j,t}$; differenced IIS (or zero-sum pairwise IIS) for an indicator $Z_{it}$, defined as $\Delta I_{it}$; many many variables for a set of $K$ potential regressors $\{x_{kt}, \ k = 1, \ldots, K\}$ for $K > T$; factors; principal components; and multiplicative indicator saturation for the set of $S_{it}x_{kt}$; see Ericsson (2011b). In related extensions, Pretis et al. (2016) propose and apply designer breaks to detect volcanic eruptions from temperature data; and Castle et al. (2020) develop path indicator saturation (PathIS) for making medium-term forecasts of COVID-19 confirmed cases and deaths. See Ericsson (2011b, 2012), Castle et al. (2013), and Section 6 for details, discussion, and examples in the literature.

The chosen saturation procedure may itself be a combination of extensions; and that choice may affect the power of the procedure to detect specific alternatives. For instance, in Example 2 above, the 37 impulse indicators $\{I_{it}, \ i = 64, \ldots, 100\}$ are not a particularly parsimonious way of expressing the step shift that occurs two thirds of the way through the sample, whereas the single one-off step dummy $S_{64t}$ is. That is, a step dummy can be a

convenient way of capturing a persistent shift in the mean. In the current context, super saturation (searching across both impulses and steps) is thus of particular interest.

*Implementation.* Because of the computational and statistical aspects associated with indicator saturation, software implementation is critical; and software packages with indicator saturation techniques are readily available. Doornik and Hendry's (2018) Autometrics includes IIS, SIS, trend indicator saturation (TIS), differenced IIS (DIIS), and their combinations, with multiplicative indicator saturation, PathIS, and designer breaks easily added by the modeler. Pretis et al.'s (2018) R package "GETS" implements IIS, SIS, TIS, and designer breaks. IHS Global 's (2022) EViews version 12 also includes IIS. See Campos et al. (2005a, 2005b) and Hendry and Doornik (2014) for the intuition of and statistical underpinnings to the general-to-specific (aka GETS) approach to modeling.

Algorithmically, IIS also solves the problem of having more potential regressors than observations by testing and selecting over blocks of variables. That block approach permits testing the aggregation assumption implied by the use of foreign aggregates in a GVAR; see Ericsson (2011a) and Hendry and Doornik (2014) for discussion of the underlying theory and for the implementation in Autometrics.

As a more general observation, different types of indicators are adept at characterizing different sorts of breaks: impulse dummies $\{I_{it}\}$ for date-specific anomalies, step dummies $\{S_{it}\}$ for level shifts, and broken trends $\{T_{it}\}$ for evolving developments. Transformations of the variable being analyzed also may affect the interpretation of the retained indicators. For instance, an impulse dummy for a growth rate implies a level shift for the (log) level of the variable.

Saturation-based tests can serve both as diagnostic tools to detect what is wrong with the model, and as developmental tools to suggest how the model can be improved. Clearly, "rejection of the null does not imply the alternative". However, for time series data, the date-specific nature of saturation procedures can aid in identifying important sources of model mis-specification. Use of these tests in model development and design is consistent with a progressive modeling approach; see White (1990); Hendry and Doornik (2014); and Hendry and Johansen (2015). Hendry (2015) and Castle and Hendry (2019) show at an intuitive level how these tools are accessible for empirical macro-econometric modeling of economic time series.

## 3. Background

Mauritania is located in the arid Sahel region of West Africa, with the Atlantic Ocean as part of its western border; see Figure 5. The population is concentrated along the southern border next to Senegal, as most of the rest of the land in Mauritania is desert and unable to sustain much life.

Mauritania is one of the world's least economically developed countries. It has a population of approximately three million, and United Nations Development Programme (2015) ranks Mauritania as 156 ("Low Human Development") out of 188 in the UN Human Development Index. Poverty affects 68 percent of rural inhabitants, who are highly vulnerable to food insecurity. A food deficit is persistent: the means of production are limited, agricultural capacity is under-exploited, and the existing farmland is under threat from desertification. Years of low agricultural output (e.g., 30 percent of national cereal needs) have led to a high dependence on imports and food aid. For more than a decade, Mauritania has been dependent on food from the UN World Food Program due to droughts, repeated food crises, political instability, and lawlessness, especially in the east and north of the country. According to UNICEF, the level of malnutrition in the country remains high: acute malnutrition is 12.6 percent nation-wide and exceeds 15 percent in some regions of the country.

Like many other economically poor countries, Mauritania is largely dependent on agriculture. Two main farming methods are practiced in Mauritania: dieri and oualo. Dieri is completely dependent on rainfall; oualo is dependent on the annual flooding of the Senegal River and Gorgol River, which again depends on rainfall. Farmers typically require

400–500 mm annually for dieri cultivation; the main wet months are from June or July through September or October.

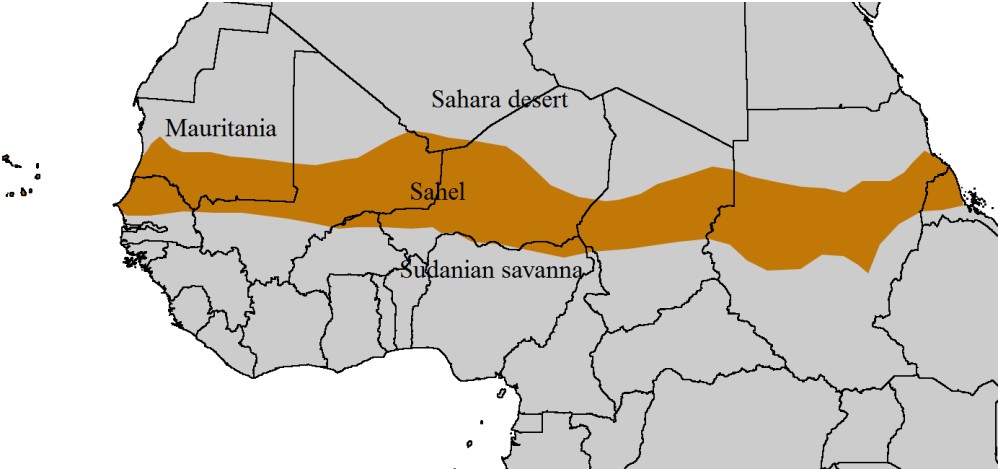

**Figure 5.** The Sahel Region in Africa (Source for map: Felix Koenig and T.L. Miles, WikiMedia Commons, commons.wikimedia.org/wiki/File:Sahel_Map-Africa_rough.png#, with added labels; accessed on 8 November 2022).

Since the 1960s, the increased frequency of droughts has forced the population to migrate towards its southern border with Senegal. Severe droughts in the Sahel region occurred during 1968–1974 and 1983–1985. In Mauritania specifically, those droughts affected food production dramatically. The country could cover only one third of its grain needs during the drought of 1968–1974; and food production declined to 3–8 percent of the grain needs during 1983–1985; see Handloff (1990). With that in mind, Section 4 below examines available data on drought and rainfall in Mauritania.

It is also valuable to consider this background in a regional and global perspective. The Sahel as a whole is suffering from the adverse effects of global climate change; see Dai et al. (2004). Dore (2005) reviews the literature on the globally changing pattern of observed precipitation at major regional and continental levels. Dore's survey reaches several conclusions. (a) Increases in the variance of precipitation are widespread. (b) Consistent with that increased variance, wet areas are becoming wetter, and dry and arid areas are becoming drier and more arid. (c) Precipitation is increasing in high latitudes (Northern Hemisphere). (d) Precipitation is declining in China, Australia, and the Small Island States in the Pacific. (e) Equatorial regions show an increased variance but no definite pattern in the mean, with one exception: the Sahel has had a persistent decline in rainfall since the late 1960s.

The ability to grow food is of paramount importance, and doing so is dependent on the availability of water. As noted above, there has been a pattern of increased aridity in North Africa since the late 1960s, with that aridity being more persistent in the western regions such as Mauritania. The driest period was in the 1980s, with some increased rainfall occurring during the 1990s, particularly in the easternmost sectors of North Africa where rainfall in some years was near or just above the long-term means; see Nicholson et al. (2000). On the other hand, southern Africa was relatively moist in the 1950s and 1970s (Nicholson et al. (2000)), but Hulme (1996) found significant decreases in precipitation being observed since the late 1970s. Early 2000 saw flood-producing rains in the eastern part of southern Africa, possibly due to the effects of the El Niño–Southern Oscillation (ENSO). The distributional consequences are particularly important for the poorest countries, which are mostly in Africa. In contrast to the general increase in the mean precipitation of other continents due to climate change, some parts of Africa have suffered increasingly severe droughts in recent decades; see Ntale and Gan (2003). Since 1995, record low precipitation has been observed in equatorial regions, while the sub-tropics have recovered from their anomalously low values of the 1980s.

Demarée (1990) examined one weather station (Kaedi) in Mauritania for changes in the means of rainfall patterns. He used a Pettitt test to look for change at a location over time where there are changes in levels of the series. The test identifies 1967 as the year for a change in levels, significant at the 1% level. This is in agreement with Dore and Lamarche (2006), who found evidence of a dramatic decline in precipitation in the Sahel as a whole, enough to characterize it as a "structural break" in the distribution of precipitation. Section 5 below tests for and identifies similar breaks in measured Mauritanian rainfall.

## 4. Data

As a prelude to and motivation for considering the data on rainfall itself, Section 4.1 examines the Palmer Drought Severity Index for Mauritania. Section 4.2 then describes the underlying rainfall data available for Mauritania and the aggregated measure of rainfall to be examined. Section 4.3 considers some potential limitations of this data before turning to its econometric analysis in Section 5.

### 4.1. Palmer Drought Severity Index

Measures of drought provide complementary descriptive evidence on changes in Mauritania's rainfall. A drought is caused by an extended period of dryness. Thus, to measure drought, one needs a measure of weather conditions over an extended period of time. The Palmer Drought Severity Index (PDSI) is a standard, standardized measure of the past and present weather's cumulative effect on the severity of dryness in the current month, accounting for past and present precipitation and temperature inter alia; see Palmer (1965) and National Center for Atmospheric Research (2022). A positive PDSI indicates relative wetness; a negative PDSI indicates relative dryness. Values between $+2$ and $-2$ are "near normal"; values between $-2$ and $-3$ represent a moderate drought; values between $-3$ and $-4$ represent a severe drought; and values less than $-4$ represent extreme drought.

The National Center for Atmospheric Research (NCAR) maintains an extensive dataset on monthly PDSI using surface air temperature and rainfall data for all land in the world except Antarctica and Greenland; www.cgd.ucar.edu/cas/catalog/climind/pdsi.html (accessed on 8 November 2022) and Dai et al. (2004) provide details. The data are divided by a $2.5° \times 2.5°$ grid. In order to examine extreme conditions in Mauritania, this subsection calculates the maximum and minimum PDSI over all months within a given year for the ten $2.5° \times 2.5°$ quadrants containing Mauritanian land area. See Coumou and Rahmstorf (2012) on weather extremes more generally.

Figure 6 shows those calculations for the minimum and maximum PDSI. As a note on interpretation, a sizable negative value for the minimum PDSI does not mean a drought in the whole country. Rather, it is the extreme minimum of the PDSI, across the grid and over all months in the year. Unsurprisingly, the values of the minimum PDSI and maximum PDSI for 1973 indicate a severe drought in that year. The maximum PDSI (equal to $-1.78$) is negative and nearly $-2$, and the minimum PDSI is $-6.43$. In fact, nearly every quadrant in nearly every month for 1973 had a PDSI that was more negative than $-2$. The minimum PDSIs for 1972 and 1973 are the most negative of any values in several decades, and the maximum PDSI drops sharply around 1970. Both features reflect the structural change in rainfall patterns detected in Section 5.

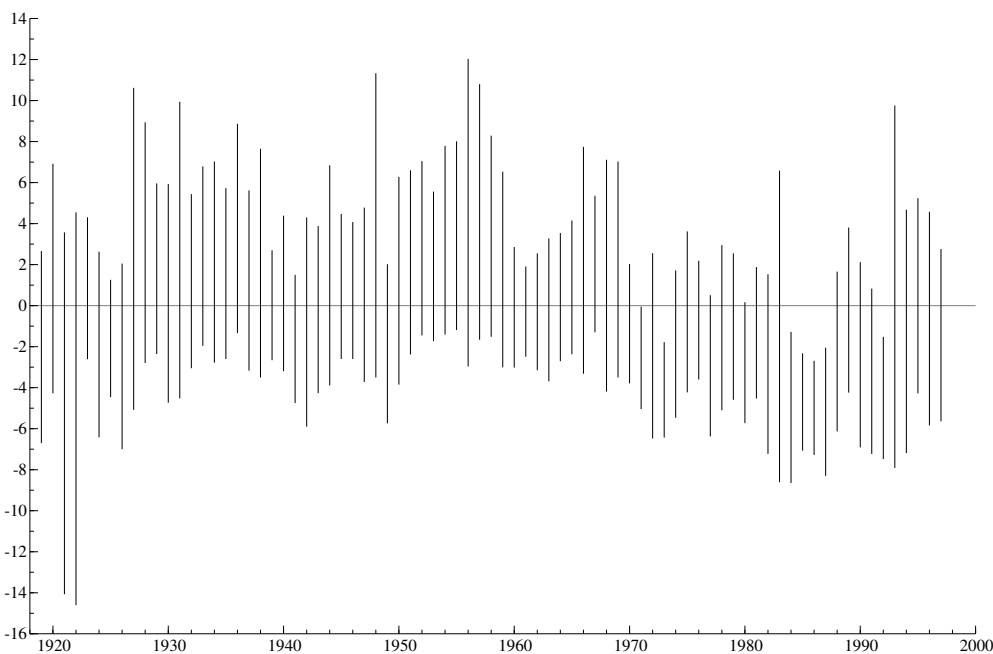

**Figure 6.** The minimum and maximum of the Palmer Drought Severity Index for Mauritania.

### 4.2. Rainfall Data

Rainfall—or the lack thereof—constitutes a key aspect of drought, so the remainder of this paper focuses on rainfall. The underlying data are monthly series for 18 raingauge stations in Mauritania, collected over 1905–2011. These data are maintained by the National Centers for Environmental Information (NCEI), which includes the former National Climatic Data Center (NCDC) and is a component of the National Oceanic and Atmospheric Administration (NOAA) in the U.S. Department of Commerce; see Peterson and Vose (1997) and Jones and Hulme (1996). The Mauritanian data are part of the Global Historical Climate Network database (version 2) (ftp://ftp.ncdc.noaa.gov/pub/data/ghcn/v2/, accessed on 8 November 2022) and are available through a convenient interface with the KNMI Climate Explorer (climexp.knmi.nl, accessed on 8 November 2022).

Table 2 provides some descriptive statistics of the data, including the stations' locations and, for each station, the data's sample period, number of observations, average rainfall, maximum rainfall, and standard deviation of rainfall. Figure 7 is a map of Mauritania, labeled with the locations of the individual stations. As Table 2 shows, rainfall patterns vary markedly across regions, with northern locations (in the Sahara) typically having minimal rainfall and southern locations having more.

**Table 2.** Descriptive statistics on observed monthly rainfall as measured at 18 Mauritanian rain-gauge stations.

| Station's Location | Sample Period | Number of Observations | Average Monthly Rainfall (mm) | Maximum Monthly Rainfall (mm) | Standard Deviation of Monthly Rainfall (mm) |
|---|---|---|---|---|---|
| Aioun el Atrouss | 1946–2011 | 673 | 20.4 | 245 | 37.9 |
| Akjoujt | 1931–2011 | 749 | 7.0 | 131 | 17.2 |
| Aleg | 1921–1997 | 918 | 20.6 | 290 | 40.8 |
| Atar | 1921–2009 | 907 | 7.0 | 121 | 15.3 |
| Bir Moghrein | 1942–2003 | 622 | 3.3 | 74 | 10.0 |
| Boghe | 1919–1997 | 941 | 23.7 | 310 | 44.0 |
| Boutilimit | 1921–2009 | 944 | 13.6 | 200 | 28.5 |
| Chinguetti | 1931–1997 | 774 | 4.4 | 99 | 10.4 |
| F'Derik | 1938–1997 | 685 | 4.2 | 117 | 11.9 |
| Kaedi | 1905–1997 | 956 | 31.2 | 364 | 56.4 |
| Kiffa | 1922–2011 | 944 | 25.5 | 325 | 47.6 |
| Moudjeria | 1911–1997 | 955 | 16.7 | 229 | 34.2 |
| Nema | 1922–2011 | 963 | 21.1 | 205 | 35.8 |
| Nouadhibou | 1906–2011 | 1137 | 1.8 | 83 | 5.8 |
| Nouakchott | 1930–2011 | 845 | 9.7 | 191 | 23.7 |
| Rosso | 1934–2011 | 754 | 21.9 | 498 | 45.9 |
| Tichitt | 1921–1997 | 826 | 6.4 | 102 | 14.2 |
| Tidjikja | 1907–2011 | 980 | 10.8 | 173 | 22.7 |

Figure 8 graphs the monthly rainfall data for all stations, highlighting the differences in typical rainfall across stations. Figure 9 graphs the same data, but across months: on each *x*-axis, "1" denotes January, "2" denotes February, etc. Visibly, much of the rainfall occurs in June–October, which are thus denoted the "wet" months.

Figure 8 also makes apparent the lack of observations for (sometimes) extended periods for every station. Some stations reported data for only a subsample of the period 1905–2011. Additionally, some observations are simply missing. To assess when the lack of observations may be important, Figure 10 plots the number of reporting raingauge stations, month by month, over 1905–2011. Until about 1920, only a few stations are collecting data, and intermittently so, in part because of World War I. Likewise, few stations are collecting data after the late 1990s. Furthermore, the stations collecting data at the ends of the sample tend to be in more populous (and wetter) locations: hence an aggregated rainfall series could be misleading during those periods. Thus, the analysis herein is restricted to the period 1919–1997.

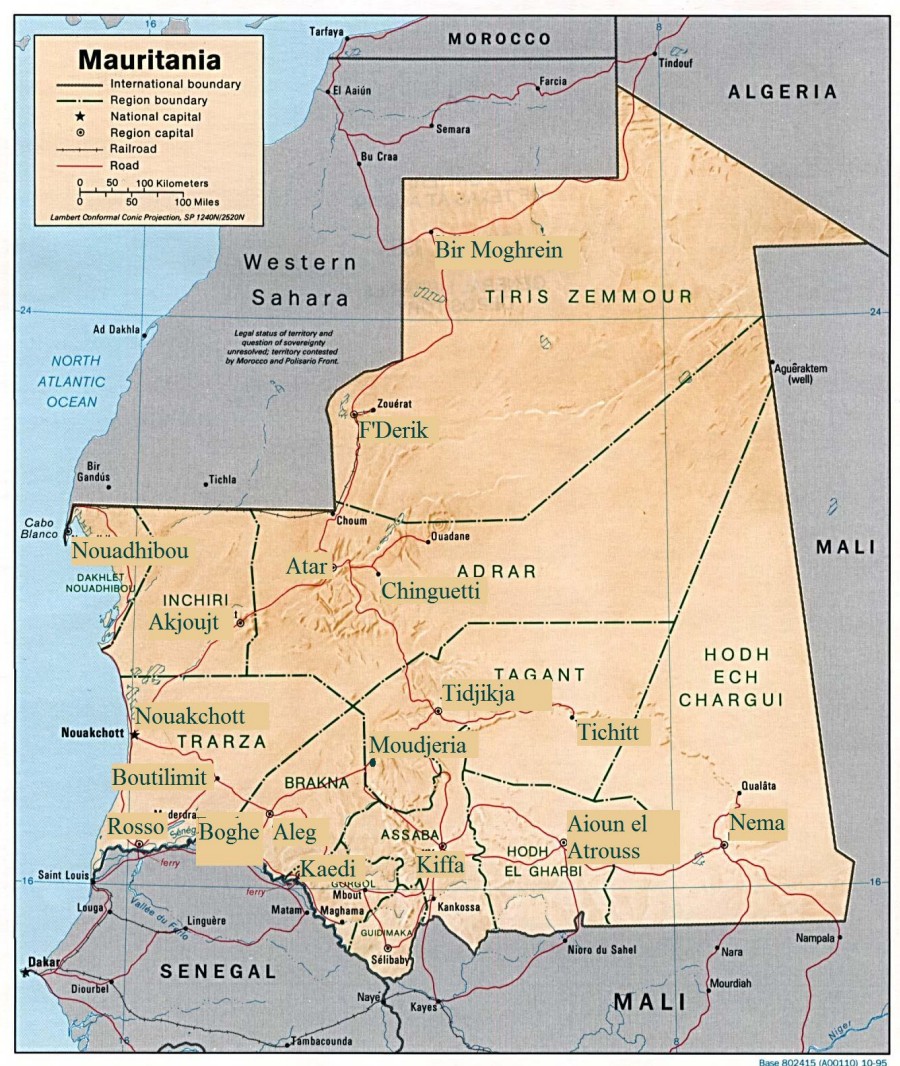

**Figure 7.** Eighteen Mauritanian raingauge stations (Source for map: the U.S. Central Intelligence Agency, available from the Perry-Castañeda Library Map Collection, University of Texas at Austin, www.lib.utexas.edu/maps/africa/mauritania_rel95.jpg, with added labels; accessed on 8 November 2022).

Figure 11 plots monthly rainfall in Mauritania for that period, where each month's rainfall is averaged across the reported rainfall for all stations reporting. If a station does not report a value for rainfall in a given month, that station is ignored in calculating the average. Two patterns are apparent in Figure 11: seasonality, which reflects the seasonality noted in Figure 9; and an apparent decline in overall rainfall, starting around 1970, with a possible modest increase towards the end of the sample. The decline from 1970 onwards is notable not only in the typical rainfall, but also in heavy rainfalls. The four wettest months are all prior to 1960; and eight months prior to 1960 have rainfall of at least 100 mm, whereas no months after 1960 do. Many years in the 1970s and 1980s have no months with more than 50 mm rainfall, whereas all but a few years before 1960 have at least one month with more than 50 mm rainfall.

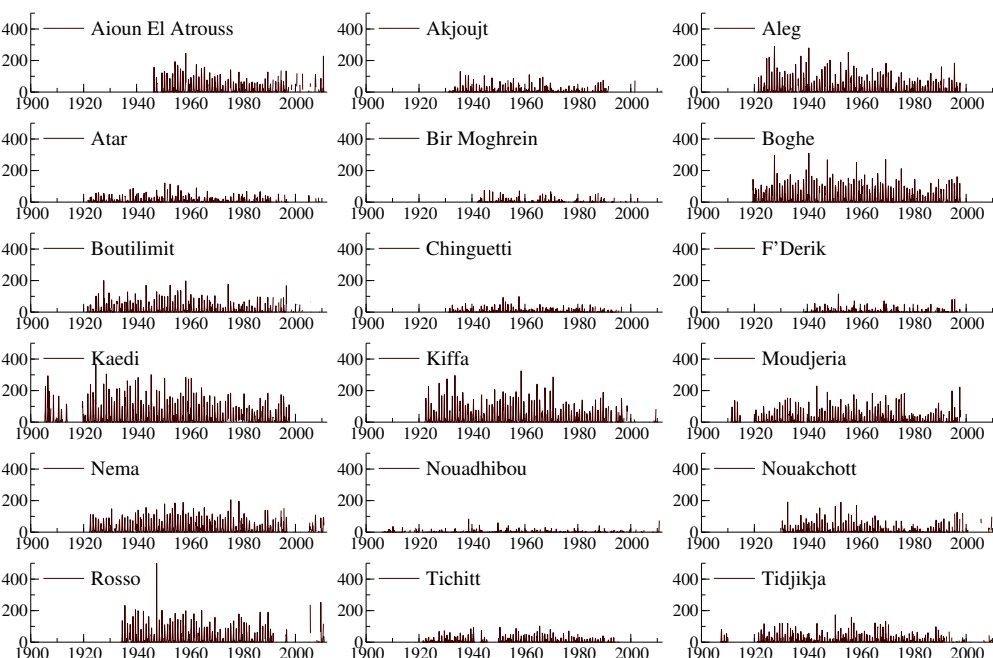

**Figure 8.** Monthly rainfall at eighteen Mauritanian raingauge stations.

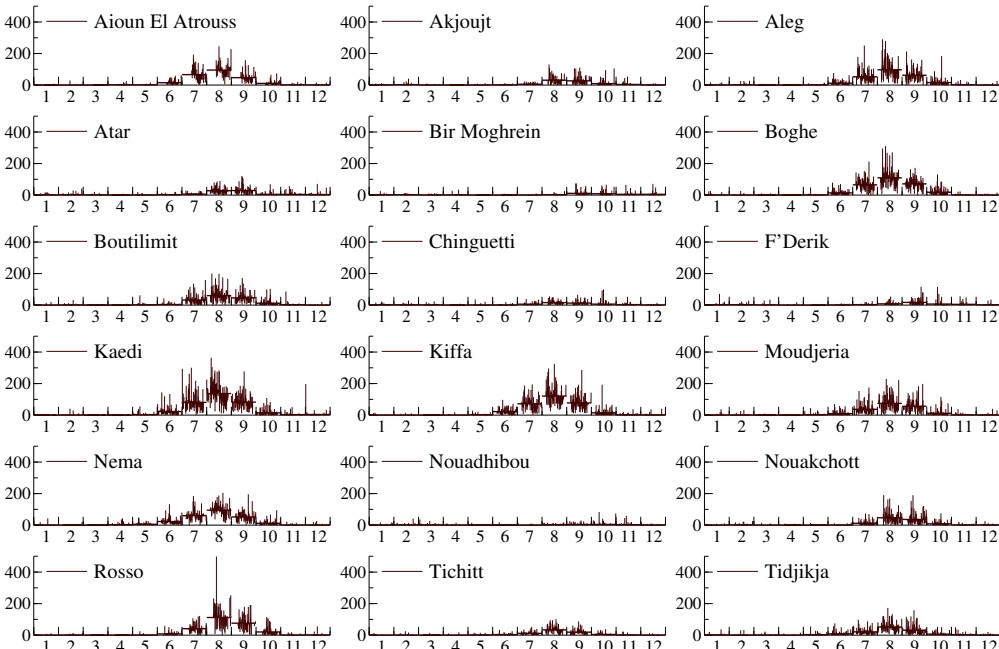

**Figure 9.** Seasonal sub-plots of monthly rainfall at eighteen Mauritanian raingauge stations.

Estimated densities provide one tool for assessing potential changes in the distribution of rainfall over time. Figure 12 plots estimated densities for low levels of monthly rainfall (0 mm–30 mm), estimated over each of the eight decades in the sample. In Figure 12, the lines thicken as the decades progress: densities for near-zero rainfall are higher in recent decades, whereas densities for rainfall in the range 12 mm–30 mm tend to be lower in recent decades. This evidence suggests an overall decline in rainfall over the sample. In addition to these graphical comparisons of the densities, statistical comparisons are feasible. However, such comparisons are not calculated here because the number of observations in each subsample is relatively small for such nonparametric comparisons, and because

the choice of decadal periods per se and of the dates spanned by each decadal period is arbitrary. Procedures in Section 5 address both these issues.

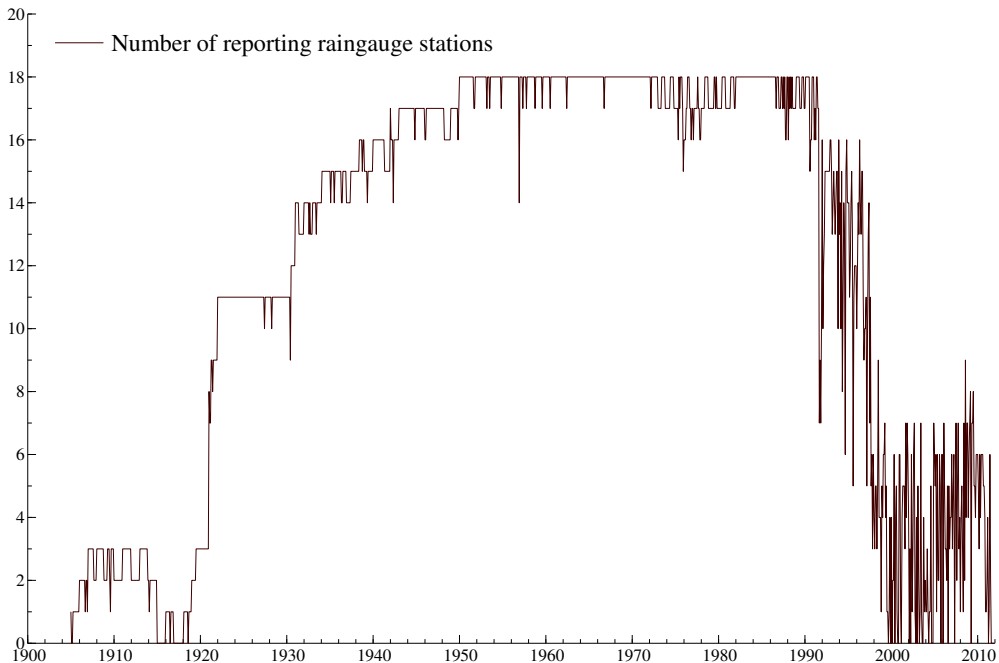

**Figure 10.** Number of reporting raingauge stations.

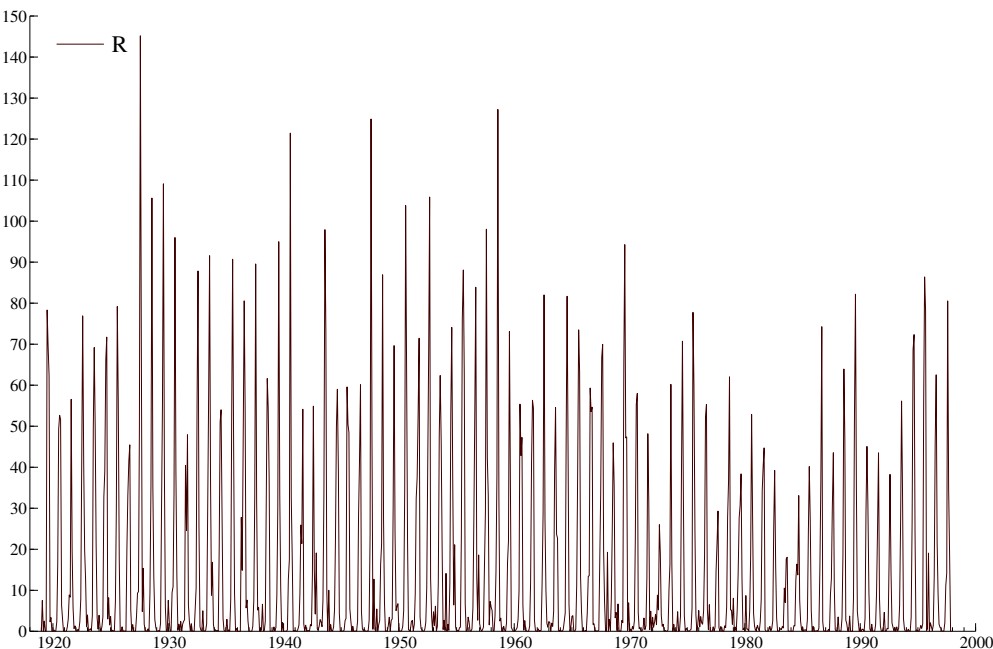

**Figure 11.** Monthly station-aggregated average rainfall, 1919–1997.

To visualize and analyze rainfall more easily, two additional adjustments to the data are made. First, only rainfall in the "wet" months (June–October) is considered. As Figure 9 highlights, very little rain falls outside those months, so little information is lost by this simplification. Also, because the wet months are the months critical for agriculture, analyzing rainfall for just the wet months is appealing economically. Second, the monthly rainfall is aggregated over time to obtain total annual rainfall. Figure 13 plots the annual rainfall for the wet months only (denoted "*RWet*") and for all months (denoted "*RTotal*").

The series exhibit very similar fluctuations and, in fact, are numerically very similar. The remainder of the analysis herein focuses on the annual rainfall for the wet months only, which is denoted $R$ for simplicity. Also, with aggregation to annual data in Figure 13, the decline in rainfall during the 1970s and 1980s is much more apparent visually.

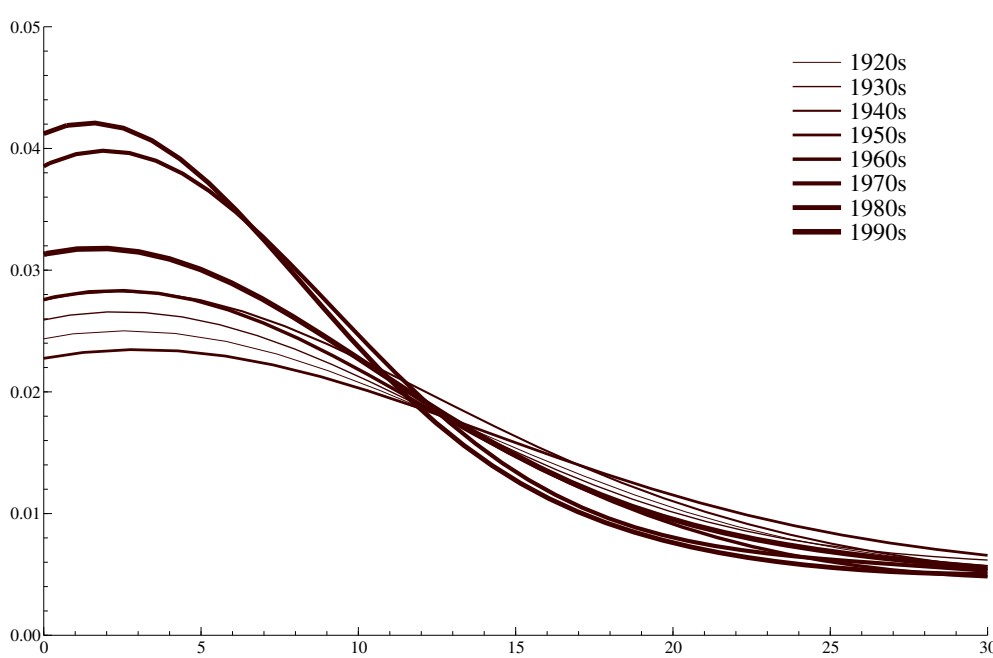

**Figure 12.** Estimated densities of station-aggregated monthly rainfall for individual decades.

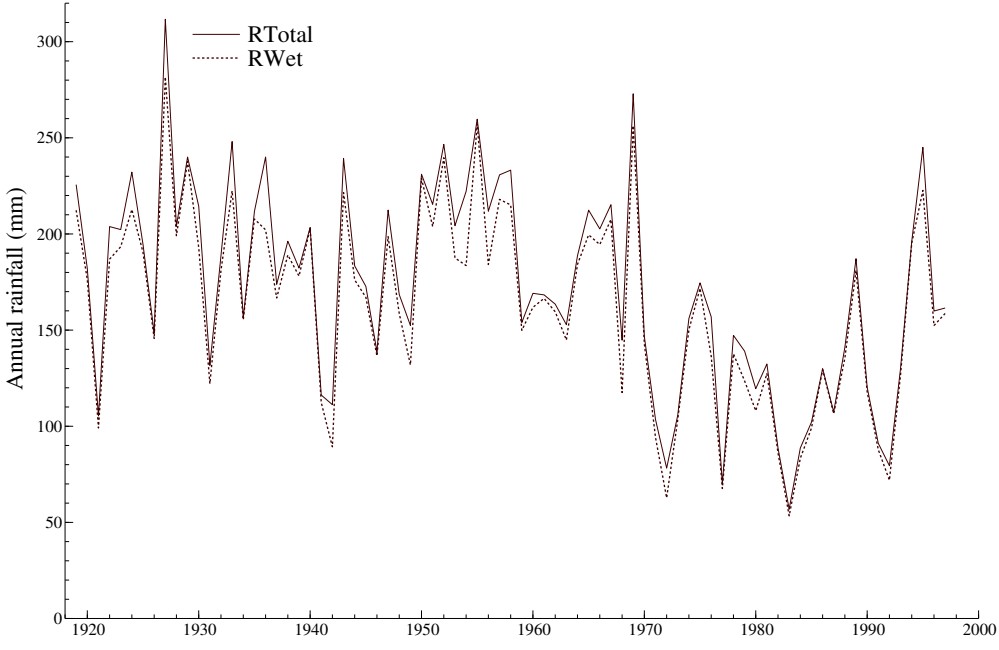

**Figure 13.** Annual station-aggregated rainfall, total (*RTotal*) and for wet months (*RWet*, or "*R*").

### 4.3. Remarks

Before turning to the detailed econometric analysis of Mauritanian rainfall, several comments about the data are worth noting. First, some monthly observations for some stations are simply missing. Whether a particular observation is missing may be "random", or not (at least to some degree). Those missing observations affect the properties of the

calculated rainfall series, which is derived from *observed* observations. Second, aggregation across raingauge stations loses information, even for months when all stations report observations. Transforming the data from (e.g.) eighteen separate observations to one "national" average loses seventeen points of data. That loss may or may not be relevant for the purposes at hand. Third, analysis of wet months ignores the information in each year's seven remaining months of data. Equivalently, analysis of the wet months treats the actual rainfall in the remaining months as if it were zero. Fourth, aggregation from monthly to annual data loses information on the monthly patterns, which may be economically important; cf. Mwale et al. (2004) and Proietti and Hillebrand (2017). Fifth, and more generally, the analysis herein makes no use of *other* time series. Important relationships may exist between rainfall and other variables such as air and sea temperatures, ENSO, air and water currents, and increased $CO_2$ emissions; cf. Hulme et al. (1998); Hendry (2011); Elliott et al. (2012); Grassi et al. (2013); and Hendry and Pretis (2013). Sixth, changes in the volatility of rainfall may be important, as well as changes in its mean—both statistically and economically. Craioveanu and Hillebrand (2012) provide an approach for analyzing level shifts in volatility. All of these comments turn on the treatment of the information in data. Hendry and Richard (1982, 1983) develop a general statistical theory for thinking about the issues involved; Campos et al. (2005b) provide a unified treatment.

## 5. Empirical Analysis

This section applies various procedures listed in Table 1 in order to assess whether a structural break has occurred in the behavior of Mauritanian rainfall and, if it has, to quantify how large that break is and when it occurred. See Doornik and Hendry (2018) for further, detailed discussion on and interpretation of the statistics themselves.

Throughout this section, the baseline model is a second-order autoregression (or "AR(2)" model) with an intercept, as reported in the second column in Table 3. This model appears reasonably well-specified from a statistical standpoint. Only one of six standard diagnostic statistics—a statistic for detecting heteroscedasticity—rejects at the 5% level; and its *p*-value is still only 2.73%. The model's estimated long-run mean rainfall is 161.3 mm with an estimated standard error of 10.4 mm. Dynamics are statistically and numerically significant, mainly through the first (i.e., one-year) lag of the dependent variable. That said, the estimated root appears stationary in that the augmented Dickey–Fuller statistic is equal to −4.08, statistically significant at the 1% level. Relative to the sample mean of rainfall, the residual standard error ($\hat{\sigma}$) is relatively large (45.70 mm), reflecting the high volatility in rainfall visible in Figure 13. As additional summary information, Figure 14 reports actual and fitted values of rainfall *R* (as time series and as a cross-plot) for the baseline AR(2) model, the residual density and histogram, the scaled residuals, and the residual autocorrelation function (ACF) and partial autocorrelation function (PACF).

This evidence on the baseline model does not, however, assess whether the estimated coefficients are constant over time. Fisher's (1922) covariance test statistic is one way of assessing constancy, but it does require specifying which subsamples to compare. Splitting the sample (arbitrarily) into one third (1921–1946) and two thirds (1947–1997), Fisher's statistic is $F(3, 71) = 2.41 \, [0.0743]$, where the numbers in parentheses are the degrees of freedom of the *F*-statistic, the number after the equality sign is the value of the *F*-statistic itself, and the number in the square brackets is the *p*-value of that *F*-statistic. Fisher's statistic does not reject the hypothesis that the coefficients in the AR(2) model are equal across these two subsamples. That said, the test is dependent upon the choice of the sample split.

Chow (1960) proposes a different approach to evaluating constancy, namely, by assessing whether the ex post forecasts generated by one subsample's coefficient estimates reasonably match the actual values in the forecast sample. One implementation of the Chow statistic utilizes recursive estimates, so it is beneficial to examine them first.

Recursive estimation of the model provides one way of dealing with the sample "split". Estimate the model over a short subsample at the beginning of the period; then sequentially add one observation at a time to the estimation sample, re-estimating over

each new subsample, until the full sample is used for estimation. Figure 15 reports the recursive least squares estimates of the first and second lags on the dependent variable $R$ and on the intercept in the baseline AR(2) model, and their recursive $t$-ratios. These do not provide a direct statistical assessment of constancy. However, there does appear to be a marked change in the coefficient on $R_{t-1}$ (the first lag of the dependent variable) and on the intercept at around 1970, with consequent changes in the corresponding $t$-ratios.

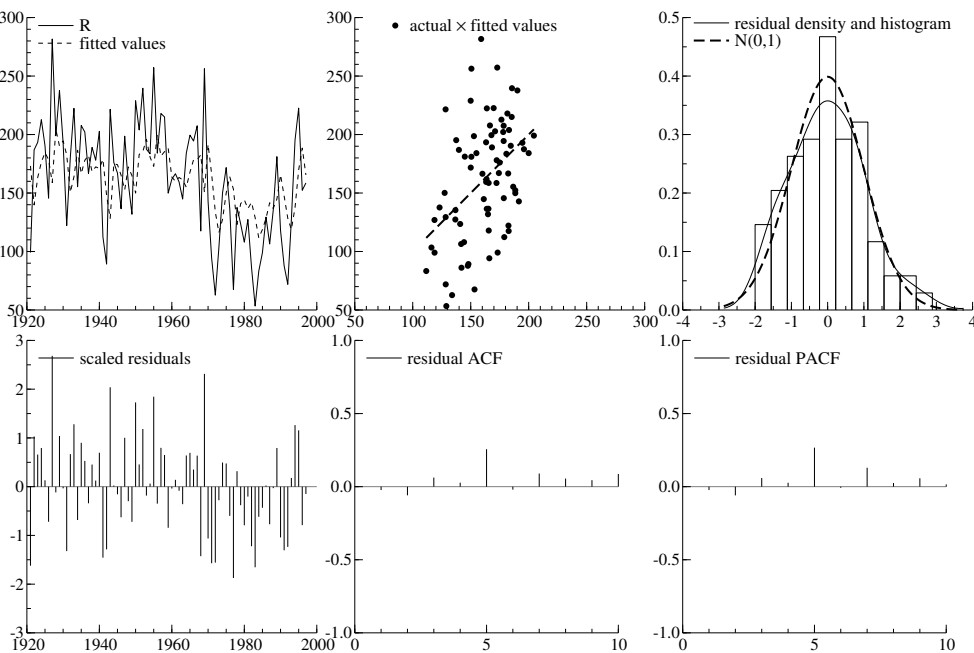

**Figure 14.** Actual and fitted values of rainfall $R$ (as time series and as a cross-plot) for the baseline AR(2) model, the residual density and histogram, the scaled residuals, and the residual ACF and PACF.

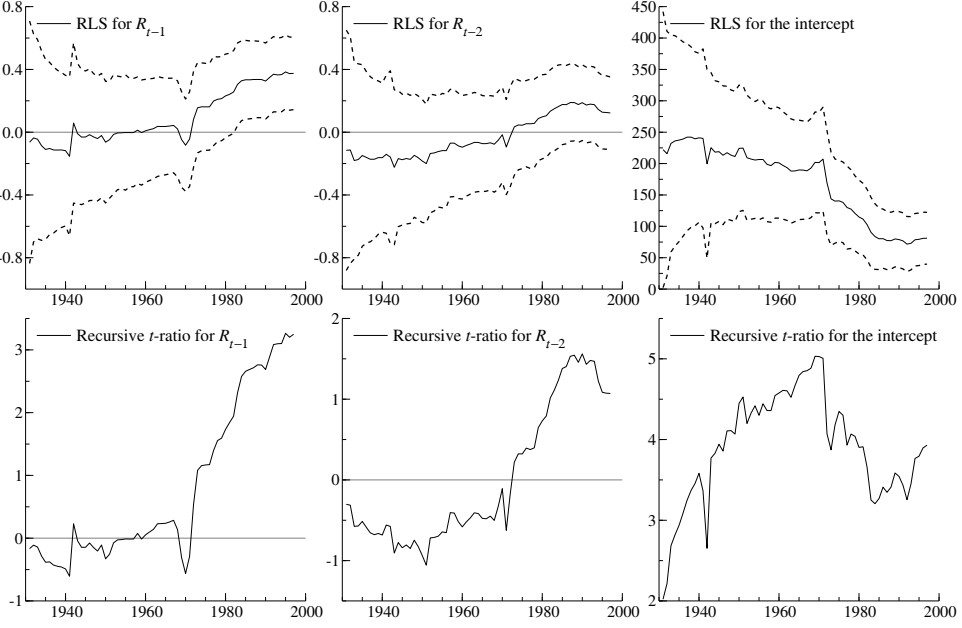

**Figure 15.** Recursive least squares estimates of the first and second lags on the dependent variable $R$ and on the intercept in the baseline AR(2) model, and their recursive $t$-ratios.

**Table 3.** Regression coefficients, estimated standard errors (in parentheses), and diagnostic statistics for several models of rainfall $R_t$.

| Regressor or Diagnostic Statistic | Model Description | | | |
|---|---|---|---|---|
| | Baseline AR(2) | Baseline AR(2), with Forecasts for 1971–1997 | AR(2) with IIS (1%) | AR(2) with Super Saturation (1%) |
| **Estimation Sample** | **1921–1997** | **1921–1970** | **1921–1997** | **1921–1997** |
| Intercept | 81.1 (20.6) | 201.5 (40.1) | 78.7 (18.9) | 191.8 (25.6) |
| $R_{t-1}$ | 0.37 (0.12) | −0.08 (0.15) | 0.44 (0.11) | 0.11 (0.10) |
| $R_{t-2}$ | 0.12 (0.11) | −0.02 (0.15) | 0.05 (0.10) | −0.16 (0.10) |
| $I_{1927t}$ | — | — | 129.1 (42.4) | 104.8 (37.1) |
| $I_{1969t}$ | — | — | 115.4 (43.0) | — |
| $S_{1970t}$ | — | — | — | −72.6 (12.7) |
| $S_{1994t}$ | — | — | — | 68.9 (20.7) |
| $\hat{\sigma}$ | 45.70 | 41.40 | 41.92 | 36.45 |
| AR(2) statistic | 1.59 [0.2118] $F(2,72)$ | 2.30 [0.1121] $F(2,45)$ | 0.96 [0.3884] $F(2,70)$ | 0.40 [0.6735] $F(2,69)$ |
| ARCH(1) statistic | 0.01 [0.9437] $F(1,75)$ | 0.02 [0.9003] $F(1,48)$ | 0.50 [0.4812] $F(1,75)$ | 0.15 [0.6988] $F(1,75)$ |
| Normality statistic | 1.30 [0.5218] $\chi^2(2)$ | 0.75 [0.6860] $\chi^2(2)$ | 0.87 [0.6481] $\chi^2(2)$ | 0.27 [0.8738] $\chi^2(2)$ |
| Heteroscedasticity statistic #1 | 2.48 [0.0511] $F(4,72)$ | 1.63 [0.1843] $F(4,45)$ | 1.29 [0.2824] $F(4,70)$ | 1.74 [0.1254] $F(6,69)$ |
| Heteroscedasticity statistic #2 | 2.70 [0.0273] $F(5,71)$ | 1.86 [0.1214] $F(5,44)$ | 1.21 [0.3126] $F(5,69)$ | 1.52 [0.1758] $F(7,68)$ |
| RESET statistic | 0.78 [0.4633] $F(2,72)$ | 0.29 [0.7520] $F(2,45)$ | 0.97 [0.3858] $F(2,70)$ | 0.84 [0.4361] $F(2,69)$ |

The recursive implementation of the Chow test statistic uses the recursive estimates to calculate forecast errors for different splits of the sample into an estimation period and a forecast period. Figure 16 plots the recursive residual sums of squares, one-step residuals, innovation errors, and the one-step, breakpoint, and forecast Chow statistics for the baseline AR(2) model. Both the one-step and breakpoint Chow statistics indicate a possible structural break at around 1970. However, the breakpoint Chow statistic for a break at 1970 is $F(27, 47) = 1.60\ [0.0781]$, statistically significant at the 10% level but not at the 5% level.

Figure 17 plots a range of auxiliary information for the baseline model estimated over 1921–1970 and with a forecast period of 1971–1997. Figure 17 includes the actual, fitted,

and forecast values of rainfall (as time series and as a cross-plot), the residual density and histogram, the scaled residuals and forecast errors, the forecasts and ± twice their standard errors, and the residual ACF and PACF. In the lower left panel, the forecast errors are systematically negative and yet the Chow statistic has little power to detect that—probably because the average forecast error is relatively small (about $-1.5\hat{\sigma}$), and because the structural break occurs near the end of the sample, implying a relatively short period for the break itself.

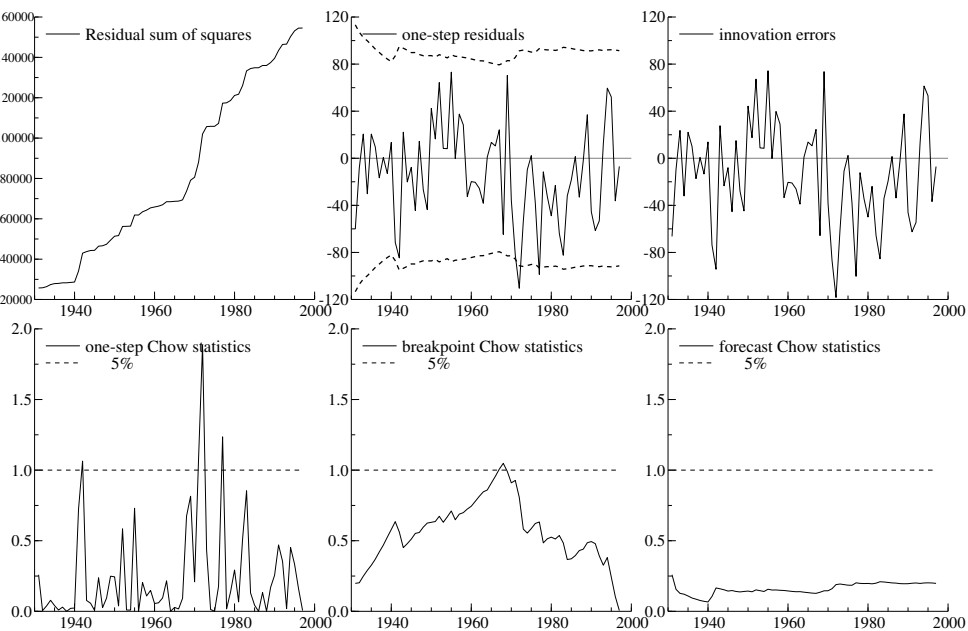

**Figure 16.** Recursive residual sums of squares, one-step residuals, innovation errors, and the one-step, breakpoint, and forecast Chow statistics for the baseline AR(2) model.

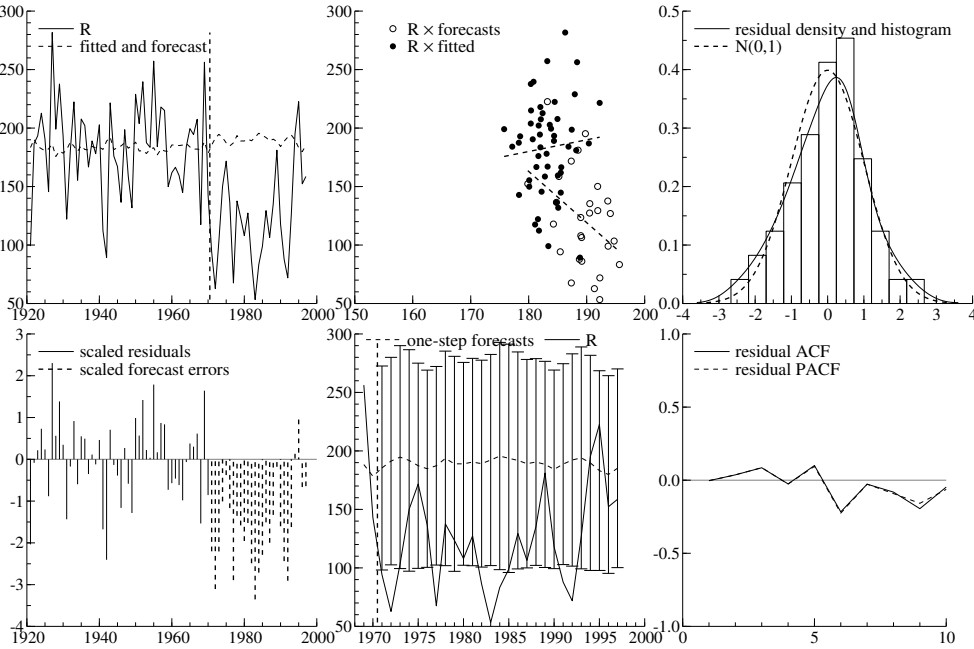

**Figure 17.** Actual, fitted, and forecast values of rainfall *R* (as time series and as a cross-plot) for the baseline AR(2) model, the residual density and histogram, the scaled residuals and forecast errors, the forecasts and ± twice their standard errors, and the residual ACF and PACF.

The middle column of Table 3 reports coefficient estimates, estimated standard errors, and diagnostic statistics for the baseline model estimated through 1970. The intercept is over twice its value from the full-sample estimate, and the autoregressive coefficients are insignificant, suggestive of rainfall being white noise through 1970. No diagnostic statistics reject.

Table 4 reports details on Bai and Perron's (1998) test statistic. Table 4 lists the Schwarz (or Bayes) information criterion for each hypothesized number of breaks, and also a modified version thereof (denoted LWZ). Both information criteria are minimized for a single break, as indicated by the values in angled brackets. Table 4 also reports the overall and conditional sup*F* statistics, which just reject at the 5% level, based on the critical values in Table 1 of Bai and Perron (1998). Conditional on there being one break, Bai and Perron's procedure estimates that the break occurs in 1970, with a 95% confidence interval of [1968, 1980]. Again, the statistical evidence is suggestive but not conclusive of a structural break around 1970.

The penultimate column in Table 3 reports the results from IIS on the baseline model at a 1% target size, with the intercept and dynamics "fixed". Two impulse indicator dummies are detected, one for 1927 and one for 1969. Both years have particularly *large* amounts of rainfall (281.7 mm and 256.3 mm respectively), as the dummies' coefficients reflect. While each dummy is highly statistically significant, neither captures the persistent fall in rainfall around 1970 that appears evident in Figure 13.

The final column in Table 3 reports the results from super saturation on the baseline model, i.e., where the machine-learning search algorithm looks at blocks taken from all impulse and step dummies at a 1% target size, with the intercept and dynamics fixed. Three dummies are detected: an impulse dummy ($I_{1927}$, as in IIS), and two step dummies ($S_{1970}$ and $S_{1994}$). Notably, dynamics are numerically and statistically insignificant, contrasting with the AR(2) model without saturation. Each step dummy is zero up until the year indicated in the dummy's name, and equal to unity from that year onward. Numerically, $S_{1970}$ captures a fall of $-72.6$ mm, over a third of the pre-1970 mean rainfall. The corresponding *t*-ratio on $S_{1970}$ is $-5.72$. That is more than "five-sigmas", which is a criterion sometimes used by experimental physicists when assessing whether a measured event is likely to have arisen purely by chance; cf. CERN (2012) and Lamb (2012). The numerical effect of the second step dummy ($S_{1994}$) nearly reverses the effect of $S_{1970}$: rainfall for 1994–1997 looks similar to pre-1970 rainfall, although having only four observations makes an accurate assessment difficult. As indicated by the diagnostic statistics in Table 3, the AR(2) baseline model appears well-specified when augmented by these three dummy variables. Figure 18 provides corresponding graphical evidence, which also (in the upper center panel) clearly separates the data into two distinct regimes.

**Table 4.** Bai and Perron's test statistics for multiple unknown breakpoints.

| Number of Breaks | Schwarz | LWZ | F($m$) | F($m \mid m-1$) |
|---|---|---|---|---|
| 0 | 7.77 | 7.90 | — | — |
| 1 | <7.59> | <7.84> | 9.96 | 9.96 |
| 2 | 7.65 | 8.03 | 6.61 | 2.60 |
| 3 | 7.73 | 8.24 | 5.34 | 2.14 |
| 4 | 7.77 | 8.42 | 5.00 | 2.72 |
| 5 | 7.82 | 8.60 | 4.78 | 2.46 |
| 6 | 7.86 | 8.78 | 4.73 | 2.58 |

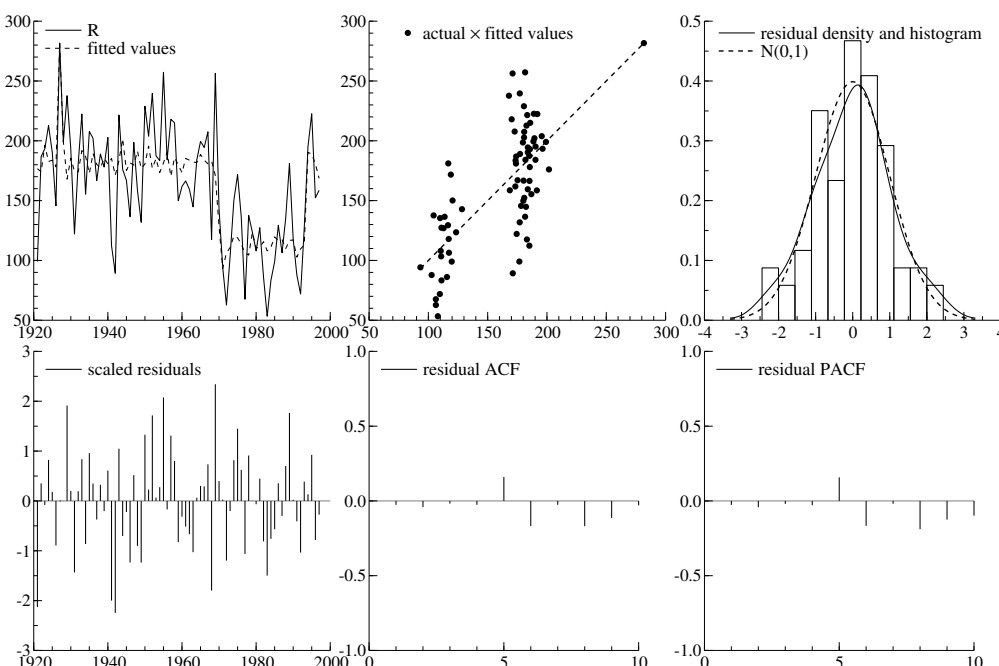

**Figure 18.** Actual and fitted values of rainfall *R* (as time series and as a cross-plot) for the modified (super-saturated) AR(2) model, the residual density and histogram, the scaled residuals, and the residual ACF and PACF.

The results in the final column of Table 3 appear robust in several directions. Other saturation approaches and other choices of target size obtain similar results. Moreover, the Chow and Bai-Perron tests also detect—albeit more weakly—the key structural break around 1970. Likewise, little is altered if *RTotal* is used rather than *RWet* (aka *R*)— unsurprising in light of Figure 13.

In summary, rainfall in Mauritania exhibits a statistically, numerically, and economically significant drop in its mean around 1970, with a possible recovery in the mid-1990s.

## 6. Remarks

Several issues merit additional remarks, including the algorithmic implementation of saturation techniques, the models considered, power, directions for further theoretical and empirical research, and potential policy implications.

First, algorithmic implementation of IIS requires important choices, as Hendry and Doornik (2014) discuss. Choices include the construction of the blocks, model selection criteria, use of diagnostic statistics, path search, block combination and re-selection, iteration, and significance level. These choices may matter under the null hypothesis of correct specification, under the alternative hypothesis, or under both.

For example, under the null hypothesis, too loose a significance level may inadvertently retain many irrelevant dummies, downwardly biasing the estimated residual standard error, and upwardly biasing *t*-ratios; see Gamber and Liebner (2017). Hendry et al. (2008) and Johansen and Nielsen (2009, 2013, 2016) consider this issue in detail. Chapter 15 in Hendry and Doornik (2014) and Johansen and Nielsen (2016) propose implementable bias corrections. Even simpler, Hendry and Doornik (2014) recommend a relatively tight significance level of $1/T$ as a rule of thumb to help keep such estimation bias minimal. In Table 3, the AR(2) model with super saturation employs a 1% level, slightly tighter than $1/T$ (=1/77). So, the indicators $\{I_{1927}, S_{1970}, S_{1994}\}$ are of substantive interest and do not appear to have been retained spuriously. Relatedly, bare-bones IIS can actually select fewer—or more—impulse indicators than Autometrics IIS, as Figure 4g,h imply.

Second, the models considered—and those not considered—can affect the model selected. Thus, the saturation results may depend on the implementation of IIS, indirectly

through which models the algorithm considers in its selection process. When the null hypothesis is false, the choice of blocks and the implied set of models can strongly influence IIS's ability to detect the alternative. Hence, Autometrics searches over many blocks, including possibly overlapping and unequally sized blocks; see Doornik (2009).

Third, IIS has power to detect many alternatives, and not just parameter nonconstancy. Applications of IIS reflect that wide-ranging ability: see Hendry (1999) on nonconstancy; Johansen and Nielsen (2009) and Marczak and Proietti (2016) on outliers; Chapter 15.6 in Hendry and Doornik (2014) on thick-tailed distributions; Hendry and Santos (2010); Gamber and Liebner (2017); and Ericsson (2017b) on heteroscedasticity; Hendry and Santos (2010) on super exogeneity; Ericsson (2011b) on omitted variables and regime changes; Castle et al. (2012) on multiple breaks; Pretis et al. (2016) on "designer" breaks; and Ericsson (2016) and Hillebrand et al. (2020) on measurement errors.

Relatedly, inclusion of other variables in the analysis may provide insights, as Section 4 discusses. Detection of indicator dummies by saturation provides a natural starting point for characterizing those indicators in a more interpretable fashion. For instance, Ericsson (2017a) detects several impulse and step indicators when analyzing forecast bias, and many of those indicators' dates correspond to turning points in the business cycle. Once dummy variables for turning points are added to the model of forecast bias, few if any indicators are detected upon re-saturation. In a similar fashion, the step shifts in rainfall detected above could be re-evaluated by adding climatic indicators such as for ENSO. That said, ENSO or similar climatic indicators appear unlikely to fully capture the detected step shifts because major ENSO events occurred before 1970 but were not detected by saturation. That implication contrasts with results in Rojas et al. (2014) and Nouaceur and Murarescu (2020). Formally incorporating ENSO or similar climatic indicators into the model with saturation could resolve these apparently conflicting results and permit an encompassing explanation.

Fourth, in order to achieve good power against many different alternatives, Hendry and Doornik (2014) intentionally allow Autometrics to beneficially (and temporarily) relax the significance level in "... search[ing] for potentially significant, but as yet omitted, variables" (p. 235). Doing so has little effect under the null hypothesis but may be helpful under alternatives.

Fifth, the persistent shifts in rainfall are much more readily detectable empirically with the 1927 indicator included in the model, even if that retained *impulse* indicator is thought of as arising purely from an "outlier". As Hendry and Doornik (2014) summarize, "[w]hen there is more than a single break, a failure to detect one [break] increases the residual variance and so lowers the probability of detecting any [other breaks]" (p. 243).

Sixth, many directions for further research on saturation techniques are highly promising. In particular, generalized saturation offers parsimonious representations of outliers and breaks; see Castle et al. (2015) and Nielsen and Qian (2022) on step indicator saturation, and Ericsson (2011b) for a typology of saturation techniques. One saturation technique—multiplicative indicator saturation—embodies a structure similar to that of regime-switching models, while allowing a given regime to differ quantitatively across its multiple occurrences. Highlighting this aspect, Table 7 in Ericsson (2017a) shows that forecast biases are not equal across different occurrences of the same "event" (or regime), where that event is a peak or a trough in the business cycle. A standard regime-switching model would have difficulty accommodating such heterogeneity, and would have difficulty even detecting turning points as regimes because of their brief nature.

Seventh, assessment of the behavior of more recent observations on Mauritanian rainfall may require a modified approach, such as analyzing the monthly station-specific data directly. Preliminary analysis along these lines has yielded promising results, using an approach similar to the econometric analysis of panel data; cf. Baltagi (1995) and Phillips (2020).

Eighth, the choice of methodology does substantively affect the ability to detect and quantify underlying changes in climate; cf. McShane and Wyner (2011). Super saturation appears particularly well-suited for capturing structural change in this time series on Mau-

ritanian rainfall. Super saturation allows inclusion of impulse indicator dummies, with an impulse dummy for 1927 appearing necessary to capture an extreme event in that year. Super saturation also allows inclusion of step dummies, which parsimoniously capture reduced average rainfall in the 1970s and 1980s and a possible subsequent increase in the mid-1990s, itself potentially arising from measurement issues. The more "standard" procedures for detecting unknown breakpoints—such as Bai and Perron's (1998) procedure—fall short in such circumstances. Such procedures have an inherent difficulty in dealing with one-period extreme events, since such events are not well-proxied by a hypothesized shift in regression coefficients that lasts several periods. Additionally, such procedures preclude detecting breaks near the end of the sample, as with the effect from the step dummy for 1994. Fortunately, impulse indicator saturation and variants thereon enhance the range of tools now available for examining structural change.

Ninth, recent research has successfully employed machine learning with saturation techniques and automatic model selection to analyze many other aspects of climate change. Hendry and Pretis (2013) examine atmospheric $CO_2$ measurements from the Mauna Loa observatory since 1958, allowing for a wide range of climatic and economic variables to determine the extent to which anthropogenic sources increase atmospheric $CO_2$. Their model controls for various natural carbon sources and sinks—such as vegetation, temperature, weather, and dynamic transport—and then ascertains the additional anthropogenic contributions from industrial production, business cycles, and shocks. Anthropogenic sources are significant contributors to changes in atmospheric $CO_2$. Castle and Hendry (2020b) model the causal roles of atmospheric $CO_2$ levels in past Ice Ages over the last 800,000 years; see also Castle and Hendry (2020a). Pretis (2020) estimates energy-balance models, interpreted as cointegrated vector autoregressions; and Jackson et al. (2021) model the interconnectedness of polar ice sheets. Saturation and automatic model selection are key to these analyses.

Behavioral aspects involving climate change are also amenable to saturation techniques. Martinez (2020) employs a multidisciplinary approach with saturation to show that, for US hurricanes since 1955, larger errors in a hurricane's predicted landfall location increase the hurricane's damages. Hendry (2020) models $CO_2$ emissions in the United Kingdom over the last century and a half, demonstrating that emissions have dropped dramatically to pre-1900 levels, even while real income increased manyfold. Legislation and technological improvements are key factors to explaining the reduction in $CO_2$ emissions. Morana and Sbrana (2019); Aldy et al. (2021); Baiardi and Morana (2021), and the special issues edited by Hillebrand et al. (2020) and Morana (2020) feature a multitude of analyses that could be extended with saturation techniques.

Finally, and at a practical level, a long-term plan for adaptation to climate change may be desirable, not only for Mauritania but for other countries in the Sahel. That said, Mauritania has had continued political instability, and the country seems to move from crisis to crisis. Under such conditions, a policy for adaptation appears unlikely to be implemented soon unless political conditions change and there is a concerted effort to help local agriculture adapt to the changing climate. In planning for such adaptation, Mauritania and other countries in the Sahel might expect some international assistance, particularly on the basis of the United Nations (1992) Framework Convention on Climate Change. Article 4.1(e) of this Convention states that all Parties shall:

> Cooperate in preparing for adaptation to the impacts of climate change; develop and elaborate appropriate and integrated plans for coastal zone management, water resources and agriculture, and for the protection and rehabilitation of areas, particularly in Africa, affected by drought and desertification, as well as floods ...

Article 4.8 contains a specific commitment of financial assistance for "... [c]ountries with areas liable to drought and desertification ...". For such assistance, the decline in Mauritanian food production since 1970 can be reasonably attributed to the decline in rainfall. Put somewhat differently, a "date" is needed to establish the start of these impacts. A country may have a case for UNFCCC-based assistance if it can establish a reasonable

benchmark for the decline in local food production. For Mauritania, the detected structural break in 1970 of the distribution of precipitation may serve as a plausible claim. That is, the adverse consequences of climate change and subsequent decline in food production could be "dated" to have started around 1970; and financial assistance could be based on this evidence. Additionally, at the 2015 UNFCCC Conference of the Parties (COP 21), 195 countries (Mauritania included) adopted the Paris Agreement on greenhouse gas emissions, mitigation, and financing from 2020. That agreement went into effect on November 4, 2016, shortly before the COP 22 in Marrakesh. More recently, the COP 27 in Sharm el-Sheikh approved a loss and damage fund to assist certain countries adversely affected by climate change.

## 7. Conclusions

Various econometric techniques have helped analyze data on Mauritanian rainfall. Our main finding is that, around 1970, the mean precipitation in Mauritania declined by about one third, relative to the mean for 1921–1969. That finding is reinforced by an analysis of the Palmer Drought Severity Index. Fitted probability density functions of rainfall also show an increase in months with little or no rainfall. Because water is a scarce resource in Mauritania, this decline in rainfall—with adverse consequences on food production—has potential economic and policy consequences. These findings support what Murray-Lee (1988) reports: "[t]he climate [of Mauritania] has altered drastically since the onset of the prolonged drought in the 1960s . . . [and] that overgrazing, deforestation, denuding of ground cover around wells, poor farming methods, and overpopulation have aggravated the drought."

**Author Contributions:** All three authors contributed equally to data curation. N.R.E. and M.H.I.D. contributed equally to all other aspects of this work. All authors have read and agreed to the published version of the manuscript.

**Funding:** This research received no external funding.

**Data Availability Statement:** The data used in this study are all publicly available from NCAR, NOAA, and the KNMI Climate Explorer, as described in Section 4.

**Acknowledgments:** The views in this paper are solely the responsibility of the authors and should not be interpreted as reflecting the views of the Board of Governors of the Federal Reserve System, the Royal Bank of Canada, or of any other person associated with the Federal Reserve System or the Royal Bank of Canada. The third author was a research assistant in the Department of Economics, Climate Change Laboratory, Brock University when this research was initially undertaken. The authors are grateful to Omaima Aljubouri, David Cox, Jurgen Doornik, David Hendry, Freja Ingelstam, Luke Jackson, Andrew Martinez, Claudio Morana, Felix Pretis, and two anonymous referees for helpful discussions and comments. All numerical results were obtained using PcGive Version 15.03, Autometrics Version 2.0a, and Ox Professional Version 8.02 in 64-bit OxMetrics Version 8.03, and WinRats 7.30: see Doornik and Hendry (2018) and Doan (2007).

**Conflicts of Interest:** The authors declare no conflict of interest.

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
