# Peer review of "Detecting and Quantifying Structural Breaks in Climate"

_econometrics, doi:10.3390/econometrics10040033_

Round 1

Reviewer 2 Report

See Report Attached.

Round 2

Reviewer 2 Report

1) It would be important to cite  Pretis, Reade and Sucarat (2018), Automated General-to-Specific (GETS) Regression Modeling and Indicator Saturation for Outliers and Structural Breaks,

https://www.jstatsoft.org/article/view/v086i03 on page 6 when listing alternative algorithms that implement IIS.

2) Given the way the Palmer index used, the discussion might flow better if the palmer index section (currently section 6) is moved either to the end of section 4 or as a separate section 5 prior to the empirical analysis as additional / alternative suggestive evidence for a structural break which can then be tested quantitatively in the empirical analysis section. This might help to ensure a continuous flow of the paper without interrupting the discussion of the results with the Empirical results section and the Remarks section. Its also not really a robustness check on the analysis and so fits better next to the data discussion section.

Author Response

Authors' response to Reviewer 2, Round 2

Reviewer 2 provided two excellent suggestions, and we have incorporated them. We are grateful for these suggestions, and they have made for a stronger and better-crafted paper. Many thanks!

1] The reference Pretis, Reade, and Succarat (2018) is now cited for providing saturation techniques (p. 8). EViews version 12 is also now cited, as it includes IIS.

2] The discussion of the Palmer Drought Severity Index has been moved from the end of the paper to a subsection in the data section (pp. 11-12) and now serves to motivate the analysis of the rainfall data.